# Positional information specifies the site of organ regeneration and not tissue maintenance in planarians

Eric M Hill[1], Christian P Petersen[1,2]*

[1]Department of Molecular Biosciences, Northwestern University, Evanston, United States; [2]Robert Lurie Comprehensive Cancer Center, Northwestern University, Evanston, United States

**Abstract** Most animals undergo homeostatic tissue maintenance, yet those capable of robust regeneration in adulthood use mechanisms significantly overlapping with homeostasis. Here we show in planarians that modulations to body-wide patterning systems shift the target site for eye regeneration while still enabling homeostasis of eyes outside this region. The uncoupling of homeostasis and regeneration, which can occur during normal positional rescaling after axis truncation, is not due to altered injury signaling or stem cell activity, nor specific to eye tissue. Rather, pre-existing tissues, which are misaligned with patterning factor expression domains, compete with properly located organs for incorporation of migratory progenitors. These observations suggest that patterning factors determine sites of organ regeneration but do not solely determine the location of tissue homeostasis. These properties provide candidate explanations for how regeneration integrates pre-existing tissues and how regenerative abilities could be lost in evolution or development without eliminating long-term tissue maintenance and repair.
DOI: https://doi.org/10.7554/eLife.33680.001

*For correspondence: christian-p-petersen@ northwestern.edu

Competing interests: The authors declare that no competing interests exist.

## Introduction

Regenerative ability in adulthood is widespread but unevenly distributed across the animal kingdom, with some species displaying high regenerative capacity while other representatives of the same phyla display a more limited capability. By contrast, the ability to maintain tissue integrity and functionality via homeostatic maintenance throughout adulthood is more common (*Poss, 2010*). Regeneration is initiated by injury, and so it involves unique inputs beyond those needed for tissue maintenance and growth, such as wound healing, injury-induced activation of proliferation, tissue re-patterning, and the integration of new and old tissues. However, beyond initial responses to injury, the processes to produce new adult tissue through homeostatic maintenance or regeneration appear to occur through substantially similar mechanisms involving the shared use of tissue progenitors and stem cells for the formation of new differentiated cells. Organisms with strong regenerative ability in many cases also undergo abundant homeostatic maintenance in the absence of injury, making them ideal systems to interrogate the requirements for these processes (*Newmark and Sánchez Alvarado, 2000*; *Elliott and Sánchez Alvarado, 2013*; *Maden et al., 2013*; *Srivastava et al., 2014*; *Rodrigo Albors et al., 2015*; *Schaible et al., 2015*; *Bodnar and Coffman, 2016*). Indeed, functional studies of gene function in highly regenerative organisms, including planarians and zebrafish, indicate that a large majority of factors required for regeneration are also required for tissue maintenance in uninjured animals (*Reddien et al., 2005*; *Whitehead et al., 2005*; *Wills et al., 2008*).

Despite the similarity between regeneration and growth programs, most animals exhibit an age-dependent reduction in the ability for de novo tissue formation without a coinciding loss of

proliferative growth or tissue maintenance. Examples of this phenomenon can be found across most metazoan phyla, including *Xenopus* limbs (*Dent, 1962*; *Slack et al., 2004*), the distal tips of mammalian digits (*Borgens, 1982*; *Reginelli et al., 1995*; *Lehoczky et al., 2011*), *Drosophila* imaginal discs (*Harris et al., 2016*), and mouse myocardial tissue (*Drenckhahn et al., 2008*; *Porrello et al., 2011*). Therefore, the mechanisms accounting for age-associated loss of regenerative capacity are unlikely to derive from generic reductions in cell proliferation or differentiation. An alternative cause of regeneration attenuation could be developmental loss of embryonic axis patterning systems, which can provide robust positional and scaling information early in embryogenesis (*Reversade and De Robertis, 2005*) but are generally not sustained into adulthood in organisms with low regenerative ability in maturity.

Adult freshwater planarians, which have a nearly unlimited ability to undergo regeneration and tissue replacement through homeostasis, use constitutive positional information as an essential upstream regulator of regeneration (*Elliott and Sánchez Alvarado, 2013*). These animals continually express patterning molecules that demarcate the main body axes and are used for regional identity determination through regeneration: Wnt and FGFRL signaling for the anteroposterior (AP) axis (*Gurley et al., 2008*; *Petersen and Reddien, 2008*; *Petersen and Reddien, 2009a*; *Gurley et al., 2010*; *Petersen and Reddien, 2011*; *Hill and Petersen, 2015*; *Lander and Petersen, 2016*; *Scimone et al., 2016*), BMP signaling for the dorsoventral (DV) axis (*Molina et al., 2007*; *Reddien et al., 2007*; *Gaviño and Reddien, 2011*; *Molina et al., 2011*), and Slit/Wnt5 signaling for the mediolateral (ML) axis (*Cebrià et al., 2007*; *Gurley et al., 2010*). Genes from these pathways are expressed mainly within cells of the body-wall musculature (*Witchley et al., 2013*) and are regionally restricted (*Lander and Petersen, 2016*; *Scimone et al., 2016*) to mark territories across each axis. Although some patterning factors are induced by injury and function early in regeneration (*Petersen and Reddien, 2009a*; *Gurley et al., 2010*; *Petersen and Reddien, 2011*; *Wenemoser et al., 2012*; *Roberts-Galbraith and Newmark, 2013*; *Wurtzel et al., 2015*), the majority are expressed in specific axial territories in uninjured animals and shift their expression domain during the regeneration process to restore missing body regions (*Petersen and Reddien, 2009a*; *Gurley et al., 2010*; *Lander and Petersen, 2016*; *Scimone et al., 2016*). Perturbations to these factors can result in tissue duplications or alterations to regional proportionality, either in regenerating animals (*Bartscherer et al., 2006*; *Owen et al., 2015*; *Lander and Petersen, 2016*; *Scimone et al., 2016*) or in animals undergoing RNAi inhibition over a period of prolonged tissue homeostasis in the absence of injury (*Hill and Petersen, 2015*; *Reuter et al., 2015*; *Lander and Petersen, 2016*; *Stückemann et al., 2017*). For example, RNAi of the Wnt inhibitor *notum* produces ectopic eyes anteriorly in the head (*Hill and Petersen, 2015*), RNAi of the Wnt gene *wnt11-6/wntA* and the Wnt receptor *fzd5/8-4* produces ectopic eyes posteriorly in the head (*Scimone et al., 2016*), and RNAi of the Wnt gene *wntP-2/wnt11-5* and the Wnt co-receptor *ptk7* produces ectopic pharynges within the tail (*Lander and Petersen, 2016*; *Scimone et al., 2016*). Some dynamic expression changes of positional control genes can occur in animals depleted of stem cells, for example, expression of *wntP-2* and *ptk7* in regenerating head fragments, suggesting that at least some patterning information is not dependent on the ability to produce of missing tissues (*Petersen and Reddien, 2009a*; *Gurley et al., 2010*; *Lander and Petersen, 2016*). Therefore, patterning factors are influential in regulating axis composition both in regeneration and in homeostatic maintenance in the absence of injury.

All mature tissues in planarians derive from a body-wide pool of adult pluripotent stem cells of the neoblast population (*Wagner et al., 2011*; *Guedelhoefer and Sánchez Alvarado, 2012*). Therefore, a compelling model to account for the robustness of both pattern restoration through regeneration and pattern maintenance through perpetual homeostasis is the use of positional cues to precisely control the differentiation and targeting of planarian neoblast stem cells for tissue production at correct locations (*Reddien, 2011*). Indeed, for the D/V axis, BMP signaling can either directly or indirectly influence the specification of neoblasts into dorsal or ventral epidermal progenitors (*Wurtzel et al., 2017*), and BMP signaling is necessary to maintain D/V axis asymmetry both in regeneration and through homeostasis (*Molina et al., 2007*; *Reddien et al., 2007*; *Gaviño and Reddien, 2011*; *Molina et al., 2011*). Likewise, Wnt signaling along the A/P axis can regulate neoblast specification in both contexts as well (*Hill and Petersen, 2015*; *Reuter et al., 2015*; *Lander and Petersen, 2016*). One expectation of this model is that the sites of organ regeneration and organ

homeostasis should be identical along the body axis even if patterning information is experimentally modified.

We investigated this model by examining the regenerative and homeostatic properties of tissue duplication phenotypes generated by pattern disruption through RNAi treatment. We focused our analysis on regeneration and maintenance of the planarian eye, a simple, well characterized, and regionally restricted organ that can be specifically removed and easily studied. Using RNAi of Wnt signaling components and surgical strategies to shift head patterning information either to the anterior or posterior, our analyses indicate that sites of organ homeostasis do not always coincide with sites of organ regeneration. These results suggest that patterning molecules have a primary function to control the location of regeneration and that mature tissue can undergo growth and tissue homeostasis through progenitor acquisition independent of more precise positional cues. Collectively, these properties could account for the integration of new and pre-existing tissues during regeneration and suggest potential mechanistic differences between tissue regeneration and tissue homeostasis.

## Results

To investigate the regenerative competency and homeostatic stability of duplicated tissue structures, we first sought to establish a reliable method for the production of duplicated organs in planarians. NOTUM is a evolutionarily conserved secreted Wnt inhibitor that deacylates Wnt ligands to prohibit binding Frizzled receptors (*Kakugawa et al., 2015*; *Zhang et al., 2015*). In planarians, *notum* is an integral regulator of anterior identity and pattern (*Petersen and Reddien, 2011*; *Hill and Petersen, 2015*). *notum*(RNAi) head fragments and uninjured animals undergo anterior shifts to axial identity to produce a set of anterior eyes located anterior to the original, pre-existing eyes (*Figure 1A*). Both the ectopic and pre-existing eyes contain a normal distribution of cell types (photoreceptor neurons expressing *opsin*, and pigment cups expressing *tyrosinase*) and enervate the brain, as seen by detecting their neuronal processes with anti-ARRESTIN staining (*Figure 1B–C*). Additionally, we tested the functionality of both sets of eyes in light avoidance assays that measure travel time away from a light source through an illuminated arena. Negative phototaxis still occurred in animals with only pre-existing eyes or only ectopic eyes. Light avoidance behavior was eliminated only when all eyes were removed, indicating the functionality of both the ectopic and pre-existing eyes to detect light (*Figure 1—figure supplement 1A–E*).

We next examined the regenerative properties of pre-existing versus supernumerary eyes generated by *notum* RNAi. Resection of normal planarian eyes results in eye regeneration over approximately 2 weeks (*Deochand et al., 2016*; *LoCascio et al., 2017*). In *notum(RNAi)* animals, resection of newly formed supernumerary eyes consistently resulted in regeneration of new eyes in the same position. However, in nearly all cases, no regeneration occurred following removal of a pre-existing eye (*Figure 1D*, *Figure 1—figure supplement 2A–B*). We were able to generate *notum(RNAi)* animals with three sets of eyes either at a low frequency from either prolonged homeostatic inhibition of *notum* or by tail removal of 4-eyed *notum(RNAi)* animals (*Figure 1—figure supplement 3A–B*). In all cases, only the most anterior eyes of such animals could regenerate after removal (*Figure 1—figure supplement 3C*). Additionally, removal of all three eyes from one side of the animal similarly resulted in regeneration of only the most anterior eye, suggesting that failure of posterior eye regeneration is not due to the presence of an anterior eye. Together these results indicate that pattern alteration by *notum* inhibition likely shifts a zone of competence for eye regeneration toward the anterior of the animal.

Based on current models of positional control in planarian regeneration, we anticipated that non-regenerative, pre-existing eyes in *notum(RNAi)* animals would eventually disappear through failed homeostasis. To examine this possibility, we monitored individual 4-eyed *notum(RNAi)* animals over an extended time (over 200 days), representing more than three times the approximate length of complete eye turnover (~60 days; [*Lapan and Reddien, 2012*]). Both the regenerative ectopic eyes and the non-regenerative pre-existing eyes persisted throughout the entire 200 day experiment (*Figure 2A*). The longevity of non-regenerative eyes suggested these organs could be homeostatically maintained despite their loss of regenerative ability.

To test whether non-regenerative eyes are actively maintained through stem cell activity or are instead retained as static tissue devoid of both cell gains and losses, we examined the functional

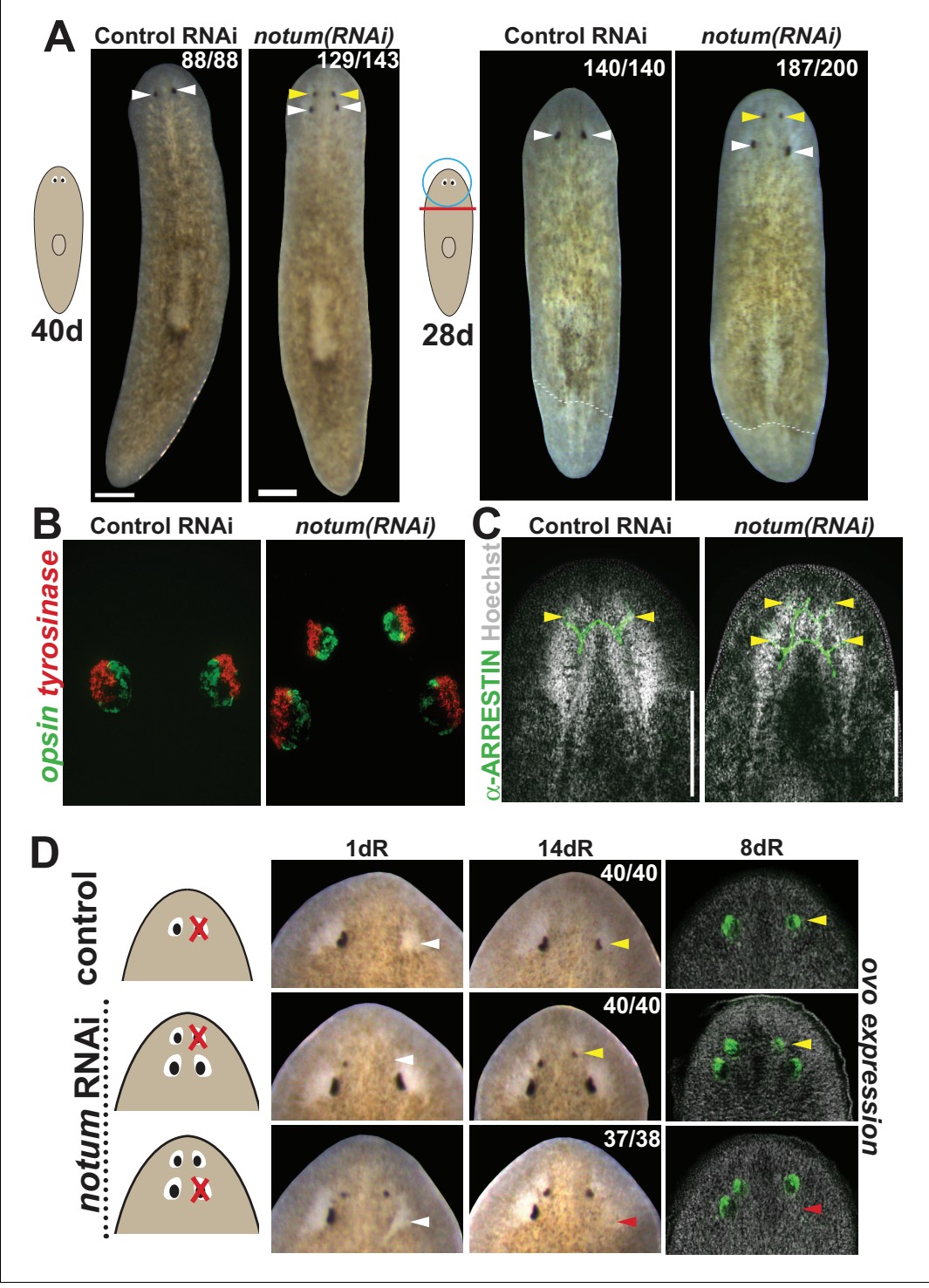

**Figure 1.** *notum* RNAi shifts the site of eye regeneration anteriorly. (**A**) Animals were treated with notum or control dsRNA every 2–3 days for (top) 40 days in the absence of injury or (bottom) for four times over 9 days followed by decapitation and 28 days of head regeneration as indicated. *notum(RNAi)* animals produced an anterior set of eyes (129/143 *notum(RNAi)* homeostasis animals and 187/200 *notum(RNAi)* regenerating head fragments, yellow arrowheads) and retained a pre-existing set of eyes (white arrowheads). (**B**) FISH to detect expression of opsin and tyrosinase. (**C**) anti-ARRESTIN immunostaining to detect photoreceptor neuron axons. (**D**) Surgical removal of eyes in control and *notum(RNAi)* animals generated by homeostatic RNAi treatment as in (**A**), showing individuals at 1 day after surgery to confirm successful removal (white arrowheads) and 14 days to assess regeneration. In *notum*

*Figure 1 continued on next page*

*Figure 1 continued*

(RNAi) animals, 40/40 anterior supernumerary eyes regenerated after removal (yellow arrowheads) and 37/38 posterior pre-existing eyes failed to regenerate (red arrowheads). Right, FISH of ovo confirms lack of eye cells produced in the region of the resected notum(RNAi) posterior eyes. Scale bars, 300 microns.

DOI: https://doi.org/10.7554/eLife.33680.002

The following figure supplements are available for figure 1:

**Figure supplement 1.** Regenerative and non-regenerative eyes both mediate negative phototaxis.
DOI: https://doi.org/10.7554/eLife.33680.003
**Figure supplement 2.** Additional controls for structure and regenerative ability of eyes from notum(RNAi) animals.
DOI: https://doi.org/10.7554/eLife.33680.004
**Figure supplement 3.** Prolonged notum RNAi and surgical strategies can create additional sets of ectopic eyes that track with regenerative ability.
DOI: https://doi.org/10.7554/eLife.33680.005

requirement of eye cell differentiation for their persistence. Four-eyed *notum(RNAi)* animals were subjected to 60 days of RNAi inhibition of *ovo*, a transcription factor that serves as a master regulator of planarian eye differentiation from neoblasts (*Lapan and Reddien, 2012*). Both the *notum (RNAi)* regenerative eyes and non-regenerative eyes disappeared with similar kinetics during *ovo* RNAi treatment, suggesting that pre-existing eyes are actively maintained through homeostasis (*Figure 2B*). To confirm these predictions, we used BrdU labeling to detect the differentiation of new eye cells. Within the eye lineage, proliferative neoblasts give rise to non-dividing eye progenitors that then terminally differentiate into mature eye cells (*Lapan and Reddien, 2011*; *Lapan and Reddien, 2012*). Therefore, the incorporation of BrdU into eye tissues allows detection of recently differentiated cells in the growing eye. We found similar numbers of BrdU +mature eye cells (*opsin +*) within both the regenerative and non-regenerative *notum(RNAi)* eyes 7 and 14 days after BrdU pulsing (*Figure 2C*, *Figure 2—figure supplement 1*), indicating that both regenerative and non-regenerative eyes are maintained by stem cell activity. BrdU incorporation was lower in each of the *notum(RNAi)* eyes compared to control eyes, but the total number of BrdU +eye cells was not altered by *notum* RNAi, suggesting that in such animals differentiating eye cells are partitioned across multiple eyes. Furthermore, we observed that both regenerative and non-regenerative eyes in *notum(RNAi)* animals undergo significant size increases in response to animal feeding (*Figure 2— figure supplement 2*). Together, these results indicate that pattern alteration through inhibition of *notum* shifted the location of regeneration but not the location of eye tissue maintenance.

We next tested whether the region of the pre-existing eye might be deficient in expression of wound-induced genes, which would provide a candidate explanation for why these organs cannot regenerate. Expression of the early wound-induced factors *jun-1* and *fos-1* as well as the late factor *gpc-1* appeared normal after resection of either anterior or posterior *notum(RNAi)* eyes (*Figure 2— figure supplement 3*). Therefore, the inability of pre-existing posterior eyes to regenerate following resection is not likely due to a failure in injury responsiveness.

An alternative explanation for the inability of pre-existing *notum(RNAi)* eye to regenerate could be positional discrepancies with respect to the rest of the body. We next examined the position of the regenerative and non-regenerative *notum(RNAi)* eyes with respect to anteriorly expressed positional control genes (PCGs) and the brain. In 4-eyed *notum(RNAi)* animals, pre-existing eyes were located far from the anteriorly expressed *sFRP-1* and within the pre-pharyngeal region of *ndl-3* expression, distinct from the location of normal eyes (*Figure 3A*). Consistent with these findings, non-regenerative *notum(RNAi)* eyes were located much more posterior than control eyes with respect to the primary body axis (*Figure 3B*). However, *notum(RNAi)* regenerative eyes were located somewhat more anteriorly with respect to the body axis compared to control eyes. *notum* RNAi can affect multiple aspects of patterning within the animal anterior (*Petersen and Reddien, 2011*; *Hill and Petersen, 2015*), so we hypothesized that if *notum* RNAi shifted the site of eye regeneration more anterior, then this site may be well positioned with respect to other anterior tissue such as the brain. Consistent with this hypothesis, non-regenerative eyes were considerably displaced with respect to *cintillo+* chemosensory neurons of the head (*Figure 3C*). Using Hoechst staining to demarcate the cephalic ganglia, we found that eyes typically formed at a particular location along the anterior-posterior axis of the brain (*Figure 3D*), consistent with previous reports in other

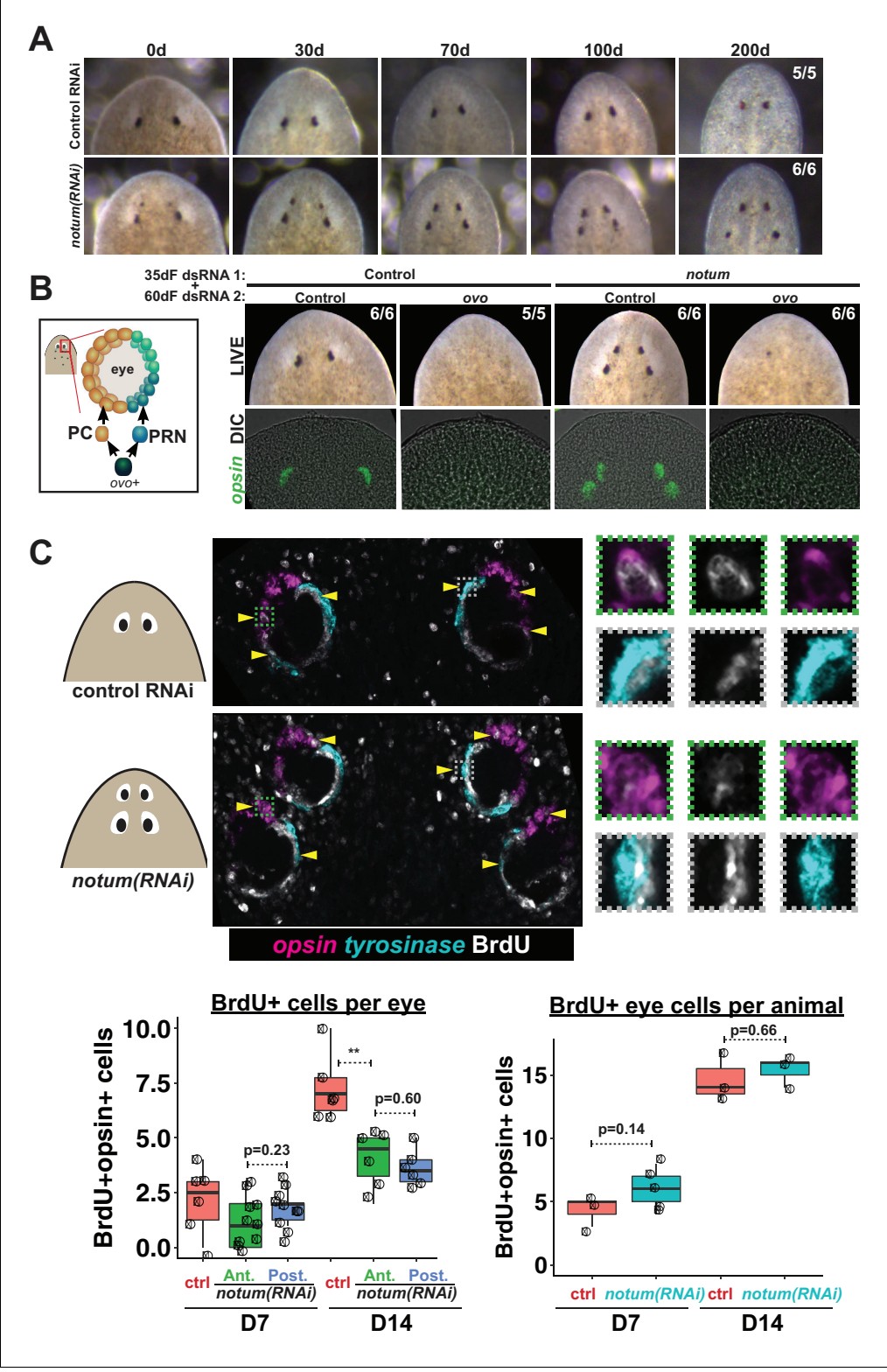

**Figure 2.** Both regenerative and non-regenerative eyes are homeostatically maintained. (**A**) Control and *notum (RNAi)* animals were fed dsRNA food every three days for 35 days then starved and individually tracked for 200 days and imaged every 30–40 days to monitor stability of the duplicated eyes. (**B**) Left, cartoon of eye differentiation showing production of photoreceptor neurons (PRN) and pigment cup cells (PC) from *ovo+* progenitors. Two-eyed control and four-eyed *notum(RNAi)* animals were generated by 35 days of dsRNA feeding

*Figure 2 continued on next page*

*Figure 2 continued*

were then treated with control or ovo dsRNA for 60 days by feeding. ovo inhibition caused loss of both the ectopic and pre-existing eyes of *notum(RNAi)* animals (12/12 sets of eyes). (**C**) Two-eyed control and four-eyed *notum(RNAi)* animals were injected with BrdU following 35 days of RNAi feeding, fixed 14 days later and stained by FISH for opsin (magenta), tyrosinase (cyan) and immunostained with anti-BrdU (gray). The head regions of BrdU-labeled notum(RNAi) animals had BrdU +cells in the anterior eyes (11/12 animals) and the posterior eyes (12/ 12 animals), a similar frequency as control animal eyes (14/14 animals). C (bottom), quantification of BrdU+ opsin+ cells after 7 or 14 days of BrdU pulsing measured per eye (left) or across all eyes (right) for each condition. p-values from 2-tailed t-tests, **p<0.01. Cartoons depict location of eyes imaged with insets showing single and multichannel enlarged images of BrdU +eye cells.

DOI: https://doi.org/10.7554/eLife.33680.006

The following figure supplements are available for figure 2:

**Figure supplement 1.** Quantification of BrdU-labeling in *notum(RNAi)* animals.
DOI: https://doi.org/10.7554/eLife.33680.007
**Figure supplement 2.** Regenerative and non-regenerative eye sizes respond to growth.
DOI: https://doi.org/10.7554/eLife.33680.008
**Figure supplement 3.** Injury-induced gene expression can occur near non-regenerative eyes.
DOI: https://doi.org/10.7554/eLife.33680.009

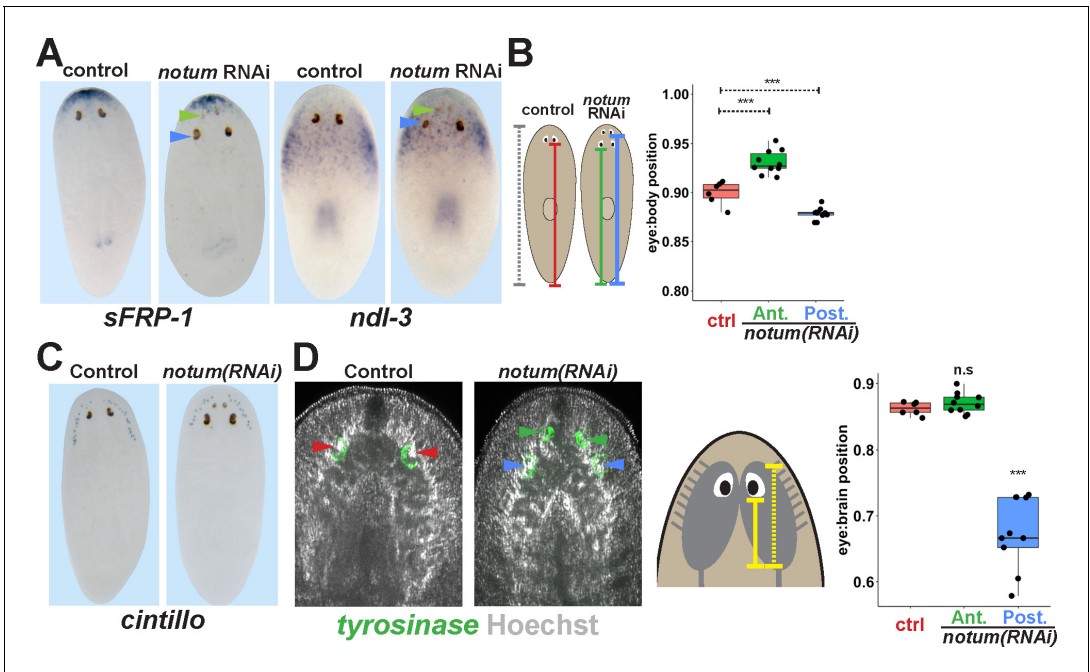

**Figure 3.** Non-regenerative eyes are mispositioned with respect to positional control genes and the brain. (**A**) WISH to detect expression of *sFRP-1* and *ndl-3* in control and *notum(RNAi)* regenerating head fragments, marking regenerative eyes (green arrows) and non-regenerative eyes (blue arrows). Posterior eyes in *notum(RNAi)* animals were located more distantly from the *sFRP-1* domain (3/3 animals) and within the *ndl-3* expression domain (6/6 animals), whereas eyes from control animals were located outside of the *ndl-3* domain (5/5 animals). (**B**) Measurement of control and *notum(RNAi)* eyes with respect to the body from fixed stained animals prepared as in (**A**). In *notum(RNAi)* animals, the supernumerary eyes are positioned more anterior and the pre-existing eyes are positioned more posterior than eyes from control animals. (**C–D**) Testing the position of eyes with respect to the brain. (**C**) Animals were prepared as in (**A**) and stained with a cintillo riboprobe labeling chemosensory neurons within a lateral territory of the head. The *notum (RNAi)* posterior eyes are located too far posterior with respect to the *cintillo* cell domain. (**D**) Measurement of the location of regenerative and non-regenerative eyes with respect to the brain, as visualized by FISH to detect tyrosinase and Hoechst staining that outlines the planarian cephalic ganglia. Right, quantifications of relative eye:brain position as determined by normalizing to the length of the brain as indicated with respect to the brain's axis. Non-regenerative eyes from *notum(RNAi)* animals (blue) have a more posterior location than eyes from control animals (red) or regenerative eyes from *notum(RNAi)* animals (green). ***, p-value<0.001 by 2-tailed t-test. n.s., p>0.05 by 2-tailed t-test.
DOI: https://doi.org/10.7554/eLife.33680.010

planarian species (*Agata et al., 1998*). Intriguingly, while *notum(RNAi)* non-regenerative eyes were located at a more posterior position with respect to the brain, regenerative eyes in *noutum(RNAi)* animals were located at the same relative position as control eyes. Therefore, the site of regeneration correlates with a particular relative location with respect to other anterior tissues, either because of a role for the brain in eye positioning or because the eye and brain are both subject to independent control by an upstream process. *notum* itself is expressed within an anterior domain of the brain in *chat+* neurons and also at the anterior pole within the body-wall musculature and both brain size and ectopic eye phenotypes from *notum* RNAi are suppressed by RNAi of *wnt11-6* (*Hill and Petersen, 2015*), which is consistent with either possibility. These observations suggest that *notum* inhibition shifted the locations of multiple tissues within the anterior, including the target site for eye regeneration, leaving behind mispositioned pre-existing eyes at a location outside of this region.

If positional control genes such as *notum* regulate the proportionality of many regional tissues, what mechanism explains the ability of non-regenerative eyes to undergo homeostatic maintenance? We considered two possible explanations for this phenomenon, either that mature eyes have an ability to induce their own progenitors in order to sustain themselves through homeostasis, or that eyes can acquire nearby eye progenitors regardless of the site of eye regeneration. Normal eye homeostasis involves migration of eye progenitors that specify from neoblasts within the anterior of the animal at a distance from the differentiated eye (*Lapan and Reddien, 2011*, *2012*). In principle, these progenitors could migrate to incorporate into either the anterior or posterior eyes of *notum(RNAi)* animals. To test this, we first examined the numbers and distribution of *ovo+* eye progenitors in 4-eyed *notum(RNAi)* animals. *notum* inhibition did not increase the number of eye progenitors per animal, and progenitors could be detected in the vicinity of both the regenerative and non-regenerative eyes (*Figure 4A*). We scored the position of eye progenitors across several animals and examined their distribution by normalizing their position to the axis defined by the head tip to the pharynx. *notum* RNAi appeared to cause a slight anterior shift to the domain of eye cell specification but that did not substantially change the abundance of eye progenitors near either the anterior or posterior eyes (*Figure 4A*). Therefore, both anterior and posterior eyes would likely have similar access to eye progenitors, consistent with the observation that the rate of BrdU incorporation into each *notum (RNAi)* eye is similar (*Figure 2C*). Furthermore, we found that nearby tissue removal, which is known to induce additional eye progenitors (*LoCascio et al., 2017*), did not enable regeneration of posterior *notum(RNAi)* eyes (*Figure 4—figure supplement 1*). Together these observations suggest that inability to regenerate is not due to a lack of access to nearby eye progenitor cells and that homeostatic maintenance of nonregenerative eyes can likely be homeostatically maintained by passively acquiring migratory eye progenitors.

The lack of increased numbers of eye progenitors or increased total BrdU eye cell labeling in 4-eyed *notum(RNAi)* animals argues against a mechanism in which differentiated eye tissue can induce eye progenitor cells. By contrast, a mechanism in which mature eyes incorporate progenitors without affecting specification predicts that non-regenerative eyes should compete with regenerative eyes for acquisition of a limited pool of eye progenitors. Consistent with this model, we found that despite generating additional eyes, *notum* inhibition did not alter total numbers of eye cells, but rather resulted reduced numbers of cells per eye (*Figure 4B*), likely due to a reallocation of the eye progenitor cell pool across an increased number of organs. To further test this model, we resected a posterior eye from 4-eyed *notum(RNAi)* animals, then counted numbers of eye cells from the ipsilateral anterior eye, using the contralateral anterior eye as an internal control. After 16 days of recovery, the posterior eye did not regenerate, as seen previously, but the anterior eye on the side of injury grew substantially larger than its contralateral counterpart (*Figure 4C*, top). Likewise, when we resected an anterior eye, the ipsilateral posterior eye enlarged compared to its contralateral counterpart (*Figure 4C*, bottom), through the size of this effect was smaller, likely due to the ability for the anterior eye to regenerate. We confirmed prior observations that removal of an eye does not substantially alter the number of *ovo+* eye progenitors on injured versus uninjured sides of the body (*LoCascio et al., 2017*), both in control and *notum(RNAi)* animals (*Figure 4—figure supplement 2*). We interpret these experiments to mean that eyes can compete with each other for acquisition of a limited pool of migratory eye progenitor cells. Together, these observations suggest that mature eyes can incorporate migratory progenitors independent of the site of eye regeneration.

The ability for patterning alteration to uncouple the sites of regeneration and homeostasis could be a phenomenon either specific to *notum* inhibition, a property specific to eyes, or alternatively

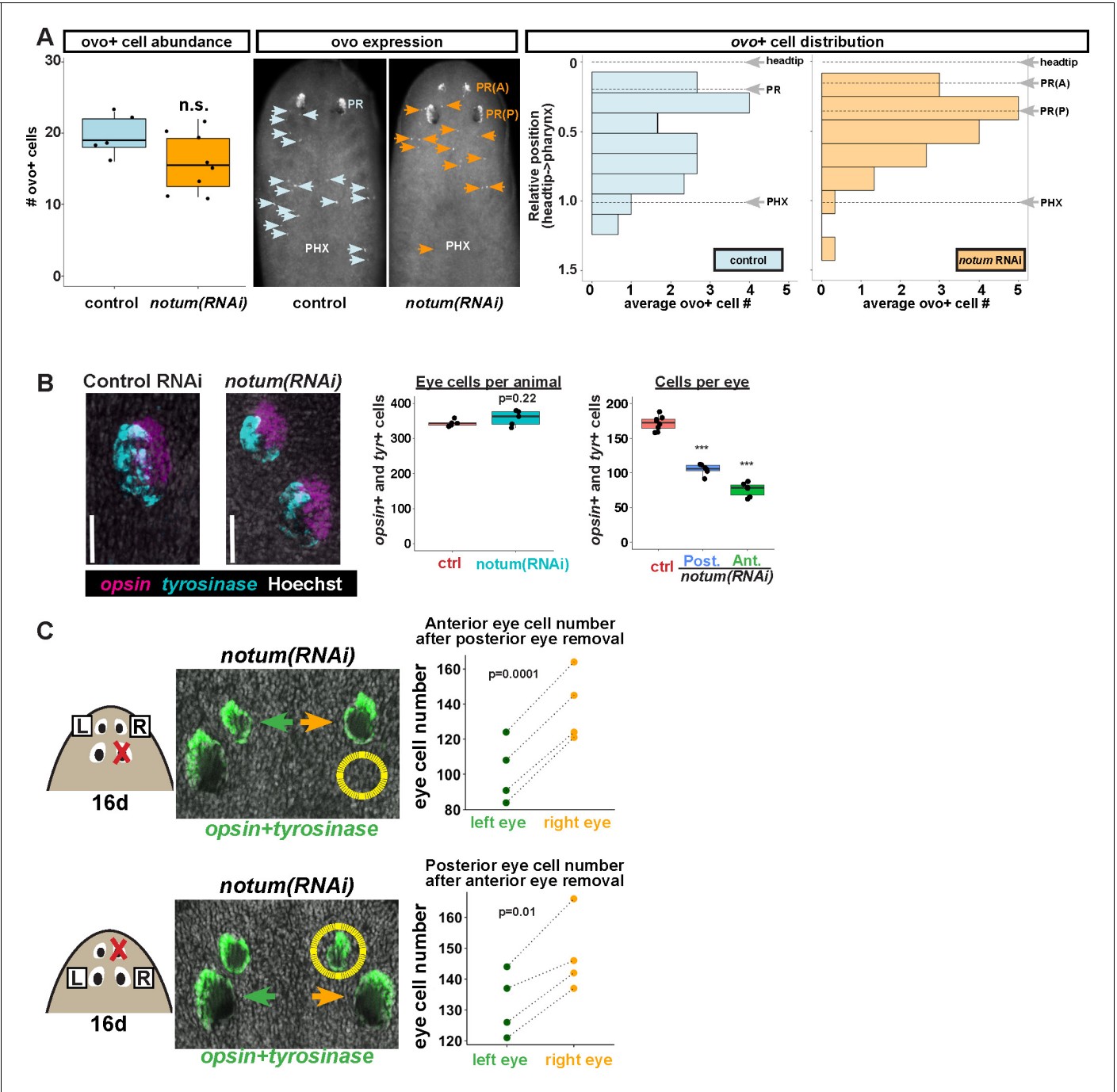

**Figure 4.** Non-regenerative eyes and regenerative eyes compete for progenitor acquisition. (A) FISH to detect *ovo+* progenitor cells located in the anterior animal region (middle panel, arrows) of control and *notum(RNAi)* animals. Left plots, *ovo+* progenitor cell numbers were not significantly altered in 4-eyed *notum(RNAi)* animals. Right plots, histograms quantifying distribution of *ovo+* eye cells showing regions anterior to the pharynx, with position normalized to the locations of the head tip and the pharynx. *notum* inhibition produced a slight anterior shift to the distribution of *ovo+* cells, but they are present in a region that includes the posterior non-regenerative eyes. (B) FISH with *opsin* and *tyrosinase* riboprobes to detect numbers of eye cells from 4-eyed *notum(RNAi)* animals and 2-eyed control animals (bars, 25 microns). Hoechst counterstaining was used to count numbers of eye cells plotted below as total eye cell numbers per animal and cells per eye. *notum* RNAi did not significantly change total eye cell numbers, and reduced the number of cells per eye. Significance determined by 2-tailed t-test, ***p<0.001. (C) Four-eyed *notum(RNAi)* animals were generated by dsRNA feeding over 40 days prior to removal of either a posterior (top) or anterior (bottom) eye on one side of the animal (R,right), leaving both eyes on the left side (L) unaffected. After 16 days of recovery, animals were fixed and stained with a combination of riboprobes for opsin and tyrosinase (green), and eye cells were quantified by counting Hoechst-positive nuclei from *opsin/tyrosinase* +cells throughout the D/V eye axis. Right, quantifications of left and right eyes from several individuals are shown and connected by dotted lines. Top, removal of a posterior eye caused the

*Figure 4 continued on next page*

Figure 4 continued

ipsilateral anterior eye (orange) to become enlarged compared to the contralateral anterior eye (green). Bottom, removal of an anterior eye caused the ipsilateral posterior eye (orange) to become enlarged compared to the contralateral posterior eye (green). Significance was measured by 2-tailed paired sample t-tests.

DOI: https://doi.org/10.7554/eLife.33680.011

The following figure supplements are available for figure 4:

**Figure supplement 1.** Effect of nearby tissue removal on posterior eye regeneration ability in *notum(RNAi)* animals.

DOI: https://doi.org/10.7554/eLife.33680.012

**Figure supplement 2.** Measurement of *ovo+* cell numbers after injury in control and *notum(RNAi)* animals.

DOI: https://doi.org/10.7554/eLife.33680.013

reflect a fundamental difference in the mechanisms of organ regeneration and homeostatic maintenance. To examine the generality of these observations, we performed similar experiments in animals after inhibition of *wnt11-6/wntA* and *fzd5/8-4* (*Figure 5A*), which act oppositely to *notum* to restrict head identity. *wnt11-6(RNAi);fzd5/8-4(RNAi)* animals form supernumerary eyes posterior to their set of pre-existing anterior eyes (*Figure 5—figure supplement 1A–B*). In these animals, ectopic posterior eyes regenerated after resection (7/10 animals), whereas pre-existing anterior eyes did not (11/11 animals), indicating that the treatment shifted the site of regeneration posteriorly (*Figure 5A*). Like *notum(RNAi)* animals, both the pre-existing and supernumerary eyes of *wnt11-6 (RNAi);fzd5/8-4(RNAi)* animals persisted for extended periods of time (*Figure 5—figure supplement 2A*) and were able to incorporate BrdU +cells through new differentiation (*Figure 5—figure supplement 2B*). These experiments verify that homeostasis can occur independent of the site of regeneration in a context other than *notum* inhibition.

To examine whether the phenomenon of shifting the site of regeneration is specific only to eyes, we focused on the pharynx, a regionalized tissue of the trunk that can be specifically removed and regenerate (*Adler et al., 2014*). *wntP-2*, *ndl-3*, or *ptk7* RNAi causes a posterior duplication of the pharynx, leaving behind a pre-existing anterior pharynx (*Sureda-Gómez et al., 2015*; *Lander and Petersen, 2016*; *Scimone et al., 2016*). In previous studies, it has been shown that the use of sodium azide to cause specific removal of both pharynges from *wntP-2(RNAi);ptk7(RNAi)* animals allowed for the regeneration of both organs (*Lander and Petersen, 2016*). However, this amputation method likely leaves behind pharynx-associated tissue such as the pharyngeal cavity and the surrounding bifurcated intestine (*Adler et al., 2014*), which could play a role in the determination of the site of pharynx regeneration. We reasoned that a broader amputation that removes this surrounding tissue would be a stronger test of the regenerative competence of these duplicated tissues. We generated animals with two pharynges after dual inhibition of *wntP-2* and *ptk7*, then performed amputations that removed either the anterior or posterior pharynx and their surrounding tissues. The ectopic posterior pharynx had almost normal capacity for regeneration while the anterior pre-existing pharynx displayed strongly diminished regenerative ability, suggesting that *wntP-2 (RNAi);ptk7(RNAi)* animals undergo a posterior shift to the site of trunk tissue regeneration (*Figure 5B*). Despite this alteration, uninjured *wntP-2(RNAi);ptk7(RNAi)* animals acquired BrdU within both pharynges after a 14 day pulse, indicating that pharynges with high or low regenerative ability both incorporate new cells homeostatically (*Figure 5—figure supplement 2C*). We conclude that modification of trunk patterning can alter the target site for pharynx regeneration away from the pre-existing pharynx without eliminating its ability to undergo homeostatic maintenance.

We additionally tested whether nou darake (*ndk*) RNAi, which produces ectopic brain tissue and ectopic eyes posteriorly into the prepharyngeal region, would similarly modify the site of eye regeneration (*Figure 5—figure supplement 3*) (*Cebrià et al., 2002*). Intriguingly, these animals displayed an unaltered site of eye regeneration, with pre-existing anterior eyes succeeding at regeneration while ectopic eyes failed to regenerate. These observations point to a distinction between the activities of *Wnt11-6/WntA* (*Kobayashi et al., 2007*) and *nou darake,* an FGFRL factor, and indicate that modification of the planarian AP axis content by RNAi does not necessarily alter the site of organ regeneration. Furthermore, these results suggest a specificity of Wnt factors in controlling the target location of organ regeneration along the primary body axis.

Finally, we tested whether the uncoupling of the site of eye regeneration and maintenance only occurs artificially after experimental gene perturbation or could occur as part of the normal

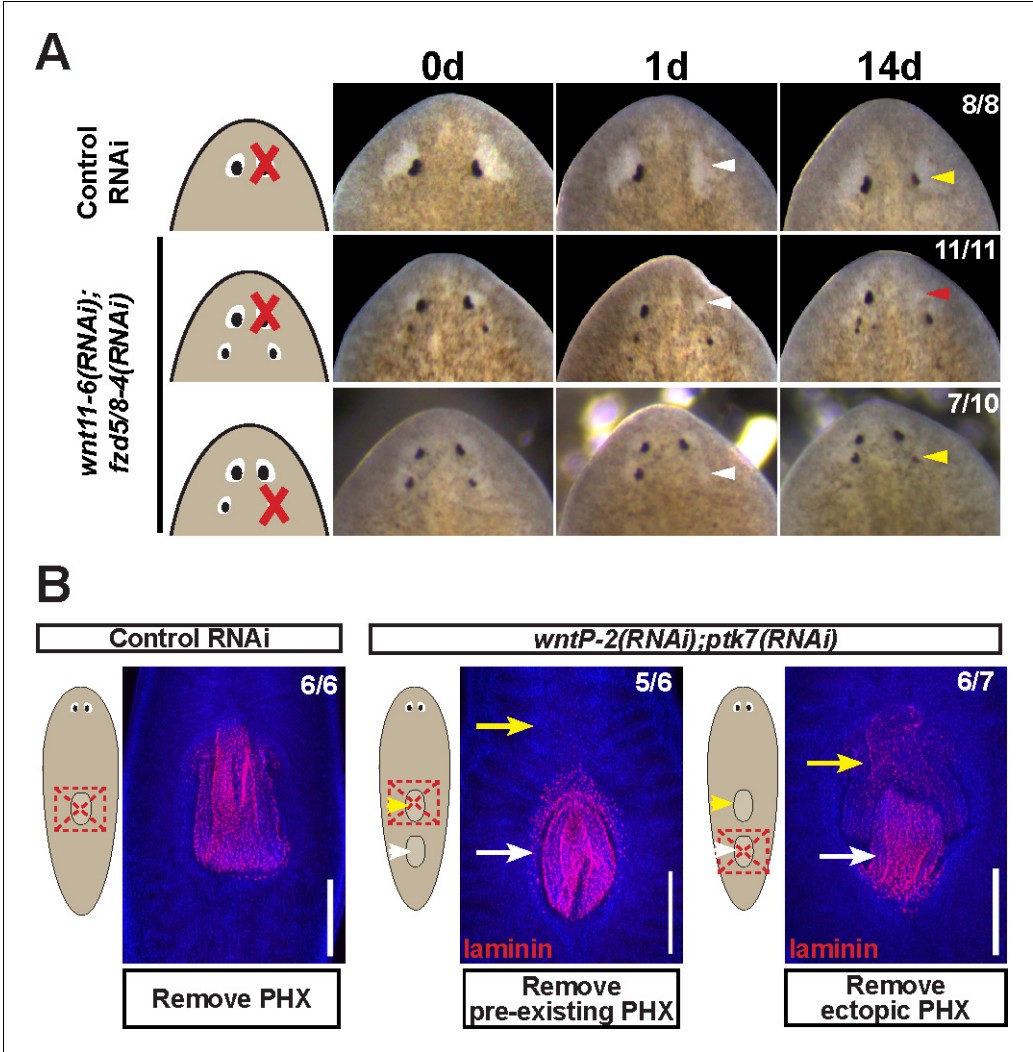

**Figure 5.** Modulation of other patterning factors alters the sites of eye or pharynx regeneration. (**A**) Simultaneous inhibition of *wnt11-6* and *fzd5/8-4* resulted in the formation of ectopic eyes posterior to the original eyes. Removal of the supernumerary, posterior eyes resulted in regeneration (7/10 animals) whereas removal of the original, anterior eyes did not result in regeneration (11/11 animals), p=0.001 by Fisher's exact test. (**B**) *wntP-2(RNAi);ptk7 (RNAi)* animals form a supernumerary posterior pharynx while retaining a pre-existing central pharynx. Cartoons denote amputations used to test regenerative ability of pre-existing or supernumerary pharynges from control or *wntP-2(RNAi);ptk7(RNAi)* animals. *wntP-2(RNAi);ptk7(RNAi)* animals were prepared by dsRNA feeding for 3 weeks, then amputated using repeated punctures centrally in a box shape around the target pharynx. Regeneration of the *wntP-2(RNAi);ptk7(RNAi)* supernumerary posterior pharynx occurred at frequencies close to those of control animal pharynges, but regeneration ability of the *wntP-2(RNAi);ptk7(RNAi)* pre-existing anterior pharynx was markedly reduced (p=0.03 by Fisher's exact test).

DOI: https://doi.org/10.7554/eLife.33680.014

The following figure supplements are available for figure 5:

**Figure supplement 1.** Additional staining and verification of the ectopic posterior eye phenotype of *wnt11-6 (RNAi);fzd5/8-RNAi(RNAi)* animals.

DOI: https://doi.org/10.7554/eLife.33680.015

**Figure supplement 2.** Tests to determine the homeostatic potential of supernumerary eyes and pharynges formed by RNAi of Wnt pathway components.

DOI: https://doi.org/10.7554/eLife.33680.016

**Figure supplement 3.** Tests to determine the regenerative potential of eyes in *ndk(RNAi)* animals.

DOI: https://doi.org/10.7554/eLife.33680.017

regenerative process. Planarians undergo a natural process of patterning alteration after amputation, in which positional control gene expression domains become altered in order to replace regional identities lost to injury as well as accommodate new reduced body proportions (*Petersen and Reddien, 2009a*; *Gurley et al., 2010*). Notably, the tissue remodeling process typically does not appear to produce intermediate states in which new well-positioned tissues are formed prior to the elimination of improperly positioned ones. However, spontaneous appearance of ectopic eyes has been reported at low frequencies, indicating errors can occur in this process (*Sakai et al., 2000*). To specifically test robustness of pattern control through remodeling, we performed a series of amputations along the primary body axis of the animal that would require an increasing amount of tissue remodeling. Our results confirmed that regenerating head fragments typically undergo remodeling through regeneration without producing a second set of eyes (*Figure 6A*). However, animals that underwent particularly severe truncations to the body axis occasionally produced supernumerary eyes during regeneration (*Figure 6A*). These results suggest that severe axis rescaling can naturally shift the putative site of eye regeneration to a location distinct from the pre-existing organ.

This observation suggested that tissue remodeling might normally involve the ability for pre-existing eyes to absorb progenitors while they are mispositioned. We hypothesized that this model would suggest that the site of eye regeneration might become distinct from the position of the pre-existing eyes during this type of regeneration. To test this, we examined the consequences of axis rescaling on the site of eye regeneration by resecting eyes from amputated head fragments in a timeseries after amputation. We fixed and stained these animals after 12 days of eye regeneration, and determined the relative location of the newly regenerated eye (*opsin+* and *tyrosinase+* cells), using the location of the midline (marked by *slit* expression) and the uninjured contralateral eye as a reference. Surprisingly, resection at early times in remodeling (days 2 and 4) resulted in eye regeneration at an anteriorly displaced position (*Figure 6B*), and these times correlate approximately with a time of dynamic alterations to patterning gene expression of *zic-1, wnt2-1, ndl-2, ndl-3* and *wnP-2* (*Figure 6—figure supplement 1*) (*Petersen and Reddien, 2008, 2009b*; *Gurley et al., 2010*; *Vásquez-Doorman and Petersen, 2014*; *Vogg et al., 2014*; *Lander and Petersen, 2016*; *Scimone et al., 2016*). Regeneration at an anterior position was dependent on complete eye removal rather than injury itself because partial resection of an eye during remodeling did not result in eye regeneration at a displaced location (*Figure 6C*).

This displacement to the site of eye regeneration eventually decayed as head fragment regeneration proceeded (*Figure 6B*), so we hypothesized that tissue remodeling might eventually realign these tissues with the target location of regeneration. To test this hypothesis, we measured the position of uninjured, pre-existing eyes versus resected, regenerating eyes with respect to the A/P axis of the brain in head fragments undergoing tissue remodeling through whole-body regeneration. Pre-existing eyes indeed gradually regained their proper position at a more anterior location with respect to the brain over several weeks of tissue remodeling (*Figure 6D*, gray). By contrast, eye removal during this process caused regeneration of a new eye located at the appropriate final position with respect to the brain (*Figure 6D*, red). Collectively, these results suggest that pre-existing tissue exerts an effect on the location of stem cell differentiation during normal tissue remodeling and can actually slow the processes by which regeneration restores proportionality, thus ensuring the maintenance of form during this transformation.

## Discussion

Our observations indicate that the location of regeneration can be altered by experimental perturbation of patterning factors or during the normal process of positional information rescaling after severe amputation (*Figure 7A–B*). Both the eyes and the pharynx use progenitors that must migrate distantly from the position where they are specified to their final differentiated location (*Lapan and Reddien, 2011*; *Lapan and Reddien, 2012*; *Adler et al., 2014*), indicating that these adult organs either can absorb progenitors that happen to encounter them or, more likely, use active trophic mechanisms for acquiring them. Our data argue that once an organ is formed, it can acquire progenitors to homeostatically maintain itself for long periods of time, perhaps indefinitely, even if it is not correctly placed with respect to patterning gene expression domains. These observations help to reconcile the fact that planarians generally regenerate perfectly, but that it is possible to recover rare variants with 'mistakes' in the process of asexual reproduction, including disorganized

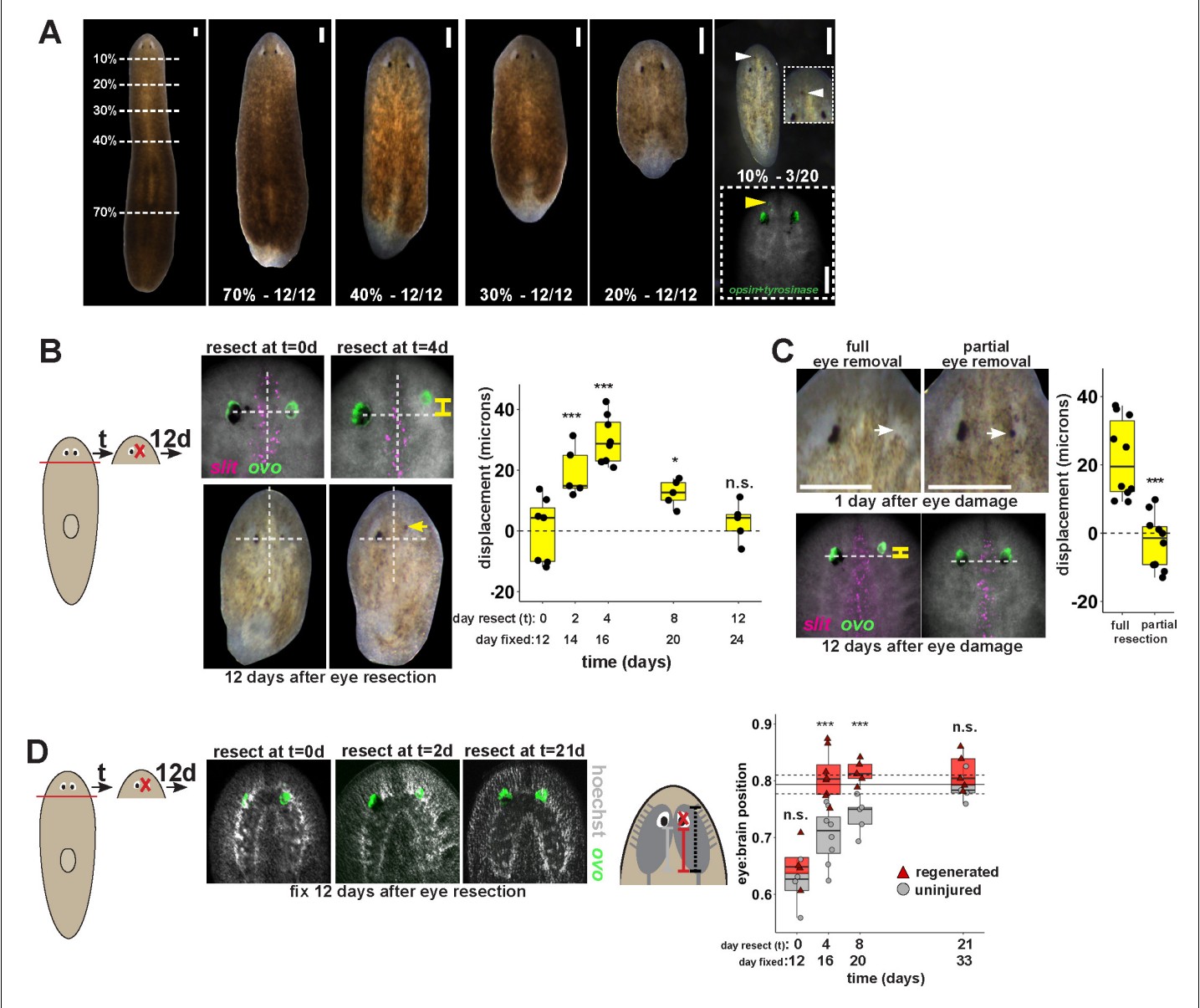

**Figure 6.** Tissue remodeling normally shifts the site of eye regeneration away from pre-existing eyes. (A) Large animals were decapitated in a series of AP positions denoted by approximate percentage of anterior tissue remaining. Such fragments regenerate into small animals that ultimately regain proportionality, and the majority of fragments had a single set of eyes throughout this tissue remodeling process (65/68 animals). However, fragments resulting from far-anterior amputations occasionally formed an ectopic set of photoreceptors during regeneration (3/20 animals). (B) Large animals were decapitated to remove ~80% of the posterior and one of the eyes within the regenerating head fragments was resected in a timeseries. Animals were fixed 12 days after eye resection and stained with an ovo riboprobe to mark the site of eye regeneration, using midline expression of slit and the A/P position of the contralateral uninjured eye as a reference (dotted lines). Right, displacement from the reference position was modified by the time of eye resection as head fragments underwent remodeling. Maximal displacement from the location of the pre-existing eye occurred when resecting eyes from d4 regenerating head fragments. (C) Tests to determine whether eye damage or eye removal is necessary for revealing the altered location of regeneration. One eye from d4 regenerating head fragments was either fully removed (left) as in (B), or only damaged to partially resect it (right). Top panels show live animals 1 day after surgery indicating successful removal versus damage to the right eye. Bottom panels show animals fixed 12 days after eye removal or damage stained and quantified for eye displacement as in (C). Only complete eye removal caused eye regeneration at an anteriorly shifted site. (D). The position of eyes from animals treated as in (B) were measured with respect to the A/P brain axis as determined by Hoechst and ovo staining. Images are projections of optical sections taken from a mid-ventral position to highlight the cephalic ganglia and dorsal positions to highlight the location of the eye. The eye:brain ratio was calculated as in *Figure 3D* by measuring the eye's distance to the posterior edge of the cephalic ganglia and normalizing to the length of the brain, with uninjured animals used to determine average eye:brain ratio at ideal proportions (solid line with dotted lines indicating standard error). Uninjured eyes successively regain proper position with respect to the brain axis as

*Figure 6 continued on next page*

*Figure 6 continued*
remodeling and regeneration proceed. Eye removal during this process results in eye regeneration at a more anteriorly displaced location that corresponds with the proper position with respect to the brain. Scale bars, 300 microns.
DOI: https://doi.org/10.7554/eLife.33680.018
The following figure supplement is available for figure 6:

**Figure supplement 1.** Expression of positional control genes is modified early during remodeling.
DOI: https://doi.org/10.7554/eLife.33680.019

supernumerary eyes that are incapable of regeneration (*Sakai et al., 2000*). Given the requirement for progenitors in organ maintenance and the inability for mature eyes to produce their own progenitors, we suggest that homeostatic eye maintenance is likely only possible within the domain occupied by *ovo+* progenitors. These interpretations suggest that patterning factors have an important role in precisely specifying the site for initiation of organ formation through regeneration but do not necessarily specify the sites of growth. We suggest that the maintenance of form in the absence of injury therefore likely involves both the use of positional control genes and the ability of existing tissues to acquire nearby progenitors for maintenance.

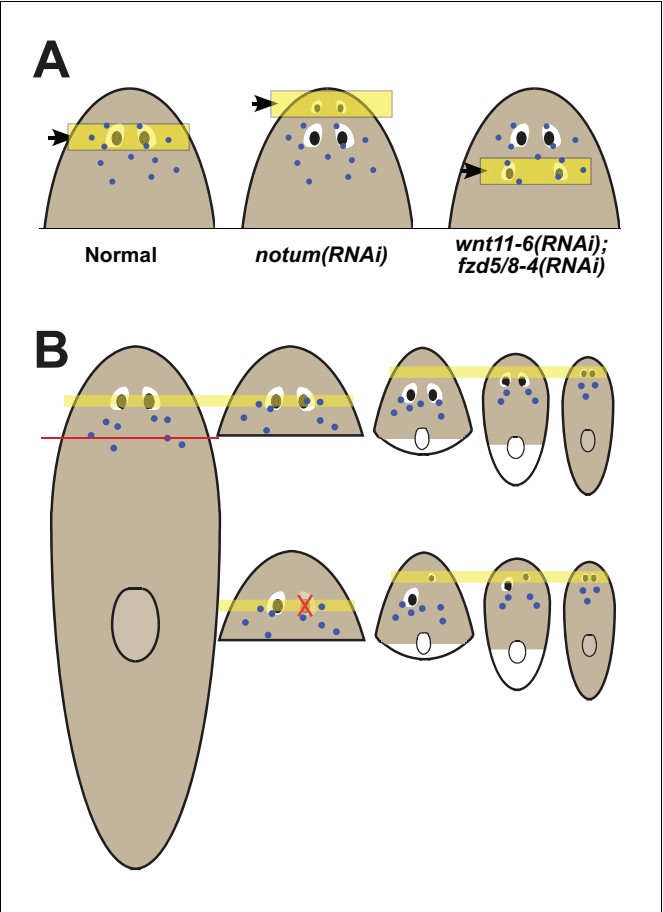

**Figure 7.** Pattern alteration uncouples the sites of regeneration and homeostasis. (**A**) Model showing shifts to the anteroposterior target site of eye regeneration (yellow box) in animals undergoing *notum* RNAi or *wnt11-6* and *fzd5/8-4* RNAi. Eye progenitors (purple dots) are present in a broader anterior domain and can renew pre-existing eyes left behind by the pattern alteration. (**B**) Shifts to the location of eye regeneration during the remodeling of head fragments (top series). Eye removal during this process results in eye regeneration at the target location for proportion re-establishment (bottom series).
DOI: https://doi.org/10.7554/eLife.33680.020

Given the ability for positional control genes to shift their domains according to the size of the new axis, a hypothetical mechanism for the restoration of form through tissue remodeling could have been new production of tissues in proper locations, followed by slow decay of old tissues in incorrect positions. However, this has generally not been observed for tissue remodeling in planarians (*Reddien and Sánchez Alvarado, 2004*). Instead, severe truncations that require extensive remodeling generally cause a slow transformation toward normal form without intermediates involving duplicated tissue (*Morgan, 1898*). The discovery that mispositioned organs can be homeostatically maintained provides a candidate model to help explain the process of tissue remodeling (sometimes called morphallaxis in planarians) (*Morgan, 1898*). Early after severe axis truncations, patterning genes dynamically shift in order to restore positional information across the body axis. New domains of progenitor specification may be defined and perhaps be mostly restored in short timescales, but pre-existing progenitors and mature tissues remain. Waves of injury-induced cell death likely accelerate the turnover of pre-existing tissues (*Pellettieri et al., 2010*), but do not appear to fully eliminate them. The ability for mispositioned organs such as eyes and pharynx to acquire progenitors, combined with the relatively slow turnover of adult organs, would then result in a gradual realignment of the positional system with respect to mature tissue. A similar process could occur in remodeling of other tissues, such as the planarian brain, or through alternate mechanisms that await discovery. The maintenance of mature tissues even when in potential conflict with the positional system could play a vital role in proper integration of new and old tissue.

Adult regenerative abilities are widespread but unevenly distributed across animal species, so an enduring question has been how these capabilities are lost or gained through evolution and organismal development. While many strongly regenerative species also maintain their tissues through ongoing tissue homeostasis, many other species maintain their tissues homeostatically without possessing strong regenerative ability as adults (*Poss, 2010*). The alteration of patterning information can be sufficient to trigger the formation of a new axis or to enhance regenerative ability in flatworm species that are refractory at head regeneration (*Liu et al., 2013*; *Sikes and Newmark, 2013*; *Umesono et al., 2013*), suggesting constitutive patterning information is vital for regeneration. Our discovery that regeneration and homeostasis can be uncoupled in a highly regenerative organism suggests a potential model for how adult regenerative ability could be lost in development or evolution. Growth and homeostatic maintenance of tissues derived from nearby progenitors would not necessarily require ongoing patterning information after axis regionalization is defined in early development. After the completion of patterning, the signaling states that enable positional gene expression could therefore be lost without sacrificing the ability for tissues to grow, be maintained, or perhaps even heal simple wounds. Whole body regeneration could have been an ancient property, as it is shared among representatives of Radiata, planarians, and acoel flatworms, with increasing evidence for common and conserved specific regulatory programs within these groups (*Srivastava et al., 2014*; *Raz et al., 2017*). Notably, these species retain patterning information constitutively during adulthood (*Reddien et al., 2007*; *Gurley et al., 2008*; *Petersen and Reddien, 2008*; *Lengfeld et al., 2009*; *Srivastava et al., 2014*; *Raz et al., 2017*), while other non-regenerative species still capable of substantial post-embryonic growth and tissue maintenance are not thought to maintain axis organization programs after embryogenesis. The loss of patterning information in adulthood therefore could account for losses of regenerative ability without the elimination of proliferation and growth.

## Materials and methods

### Planarian culture

Asexual Schmidtea mediterranea animals (CIW4 strain) were maintained in 1x Montjuic salts between 18–20°C. Animals were fed a puree of beef liver and starved for at least one week prior to the start of any experiment.

### Whole-mount in situ hybridization (WISH)

Animals were fixed and bleached as described previously (*Pearson et al., 2009*). Riboprobes (digoxigenin- or fluorescein-labeled) were synthesized by in vitro transcription (*Pearson et al., 2009*; *King and Newmark, 2013*). Antibodies were used in MABT/5% horse serum/5% Western Blocking

Reagent (Roche, Basel Switzerland) for FISH (anti-DIG-POD 1:2000 (Roche, Basel Switzerland), anti-FL-POD 1:1000 (Roche)) or NBT/BCIP WISH (anti-DIG-AP 1:4000 (Roche, Basel Switzerland)) (*King and Newmark, 2013*). For multiplex FISH, peroxidase conjugated enzyme activity was quenched between tyramide reactions by formaldehyde (4% in 1x phosphate buffered saline with 0.1% TritonX100 (PBSTx)) or sodium azide treatment (100 mM in 1xPBSTx) for at least 45 min at room temperature. Nuclear counterstaining was performed using Hoechst 33342 (Invitrogen, 1:1000 in 1xPBSTx).

## RNAi

RNAi by feeding was performed using either E. coli HT115 cultures expressing dsRNA from cDNA cloned into pPR244 (*Gurley et al., 2008*) or in vitro transcribed dsRNA (*Rouhana et al., 2013*) mixed directly into 70–80% liver paste. For head remodeling experiments, animals were fed RNAi food 4 times over 9 days and surgeries were performed on the same day as the final feeding. For long-term feeding experiments, animals were fed RNAi bacterial food every 2–3 days for the length of experiment indicated. RNAi vectors or dsRNA to inhibit *notum*, *wnt11-6*, *fzd5/8-4*, *wntP-2* and *ptk7* were described and validated previously (*Petersen and Reddien, 2011*; *Hill and Petersen, 2015*; *Lander and Petersen, 2016*)

## Whole-mount Immunostaining and BrdU Experiments

Fixations were performed by treatment with 5% N-acetyl-cysteine (NAC) in 1x phosphate buffered saline (PBS) for 5 min, 4% formaldehyde/1xPBSTx for 15 min, and bleaching overnight in 6% hydrogen peroxide in methanol on light box. Animals were blocked 6 hr in 1xPBS/0.3% TritonX-100 +0.25% bovine serum albumin (PBSTB) at room temperature. Fixed samples were allowed to incubate with primary and secondary antibodies overnight (~16 hr) at room temperature with mild agitation. ARRESTIN labeling was performed using a mouse monoclonal antibody (clone VC-1, kindly provided by R. Zayas) at 1:10,000 in PBSTB followed by incubation with anti-mouse HRP conjugated antibody (Invitrogen, 1:200 in 1xPBSTB) and tyramide amplification (Invitrogen Alexa568-TSA Kit, tyramide at final concentration of 1:150).

For BrdU labeling, two-eyed control and four-eyed *notum(RNAi)* animals were produced by 35 days of dsRNA feeding and injected with BrdU solution (5 mg/mL in water, Sigma 16880/B5002). Animals were fixed as described above 14 days after injection of BrdU. Animals were rehydrated and bleached in 6% hydrogen peroxide in PBSTx for 3–4 hr on a light box (*Thi-Kim Vu et al., 2015*). FISH was performed as described above with all HRP inactivations carried out using formaldehyde (4% in 1xPBSTx for at least 45 min). Following FISH protocol, acid hydrolyzation was performed in 2N HCl for 45 min, samples were washed with 1xPBS (twice) then 1xPBSTx (four times), and blocked in PBSTB for 6 hr at room temperature. Primary antibody incubation was performed using rat anti-BrdU antibody (1:1000 in PBSTB, Abcam 6326) overnight at room temperature, followed by 6x washes in PBSTB, and overnight incubation in anti-rat HRP secondary antibody (1:1000, Jackson ImmunoResearch 112-036-072). Tyramide development was performed at room temperature for 1 hr (Invitrogen Alexa568-TSA Kit, tyramide at final concentration of 1:150).

## Organ Specific Regeneration Assays

Worms were immobilized on a small piece of wet filter paper chilled by an aluminum block in ice. Both eyes and pharynx were resected using a hypodermic needle. For eye removal, care was taken to avoid penetrating completely through the dorsal-ventral axis of the animal. Animals were tracked individually to more accurately monitor photoreceptor regeneration. All animals were imaged one day before eye removal to establish the exact phenotype displayed, one day after eye removal to confirm removal of photoreceptor tissue, and 14 days after surgery to determine regenerative outcome. For resection of the pharynx, hypodermic needle was used to cut through the DV axis of the animal around pharynx and remove entire body region containing the organ and associated tissue from the middle of the animal. Animals were imaged both before and after pharynx removal. Pharynx regeneration was scored by in situ hybridization for the organ specific marker *laminin* (*Adler et al., 2014*).

## Light Avoidance Assay

Two-eyed and four-eyed animals were created through 40 days of control and *notum* dsRNA food respectively. Animals were given the surgeries indicated by representative images in panel C of *Figure 1—figure Supplement 1* and light avoidance was tested the following day. To test light avoidance, animals taken into a dark worm and placed a 128 mm dish across which a field of light was cast from one end (Schematic of experimental setup shown in *Figure 1—figure Supplement 1*, panel A). Animals were placed approximately 32 mm from the end of the dish closest to the light source (red circle in *Figure 1—figure Supplement 1*, panel A). Animals were then observed as they moved throughout the dish for 5 min, recording their relative distance from the light source by regional location within the dish every 30 s. Multiple worms were tested for each experimental condition shown (exact numbers shown on *Figure 1—figure Supplement 1*, panel D) and each worm was tested twice. Decapitated worms were used as an additional control and showed no response to light when placed in the dish. Any animals that begin to defecate during a trial or showed a scrunched phenotype (indicative of future defecation) were removed from the experiment. Final paths were determined as the average location of all worms of a given condition at that time point.

## Imaging

Imaging was performed with a Leica M210F dissecting scope with a Leica DFC295 camera, a Leica DM5500B compound microscope with optical sectioning by Optigrid structured illumination, Leica SP5 or Leica TCS SPE confocal compound microscopes. Fluorescent images collected by compound microscopy are maximum projections of a z-stack and adjusted for brightness and contrast using ImageJ or Adobe Photoshop.

## Relative Location, Displacement and Area measurements

Animal and brain lengths were measured with ImageJ as visualized with Hoechst. For brain length, one lobe from each animal was measured from the most posterior brain branch to the most anterior brain. In *Figures 3D* and *6D*, relative eye position with respect to the animal or brain was measured as the distance from the center of the photoreceptor as visualized by FISH for tyrosinase to the anterior animal pole or anterior most brain branch divided by total animal or brain length respectively. Eye displacement in *Figure 6B–C* was determined by the absolute difference between AP locations of individual eyes on the same animal after fixation and staining for eye cell markers (*opsin* and/or *tyrosinase*) as well as midline markers (*slit*). In *Figure 2—figure Supplement 2*, pigment cup area was measured from live images of planarian with heads fully extended using ImageJ. Changes in individual pigment cup area were measured as the area of a pigment cup following 24 days of feeding or starvation divided by the area of that same pigment cup at day 0. Samples from similar fragments and time points were averaged and significant differences determined by two-tailed Student's t-tests.

## Cell Counting

For *Figure 4A*, *ovo+* progenitor cells were identified as *ovo+* cells on the dorsal side apart from the mature eyes. Images were rotated to a common y-axis beginning at the head tip and cells were counted by manual scoring of maximum projection images in ImageJ and acquiring x-y coordinates of each scored cell. Head-tail distributions of *ovo+* cells were computed by normalizing A/P positions of the cells to the axis from a 0 to 1 range defined by the head tip to the anterior end of the pharynx as defined by Hoechst staining, then distributions determined by binning and averaging measurements from three animals for each condition. Quantification from individual animals shown as data points and potential significance was determined by two-tailed t-tests. In *Figure 4B–C*, eye cells were counted by imaging whole eyes through confocal microscopy at 63x with 0.75-micron slices and manually enumerating numbers of Hoechst + nuclei of the eye surrounded by *opsin/tyrosinase* or *ovo* FISH signal. Nuclei were manually marked within the stack and neighboring planes examined to prevent over-counting. 2-tailed paired t-tests were used to determine significance between eye cell number between injured and uninjured body sides for a series of individual animals. Cell counting experiments were performed by blind scoring.

## BrdU Colocalization Analysis

Cells showing colocalization of BrdU with markers of differentiated photoreceptor cells *opsin* or *tyrosinase* were identified manually using ImageJ from 40x magnification z-stack confocal images (0.75 micron thick slices) taken on a TCS Leica SPE confocal microscope.

## Acknowledgements

We thank R Zayas for the kind gift of the VC-1 anti-ARRESTIN antibody and members of the Petersen lab for discussions.

## Additional information

### Funding

| Funder | Grant reference number | Author |
| --- | --- | --- |
| National Institutes of Health | 2T32GM008061-31 | Eric M Hill |
| National Institutes of Health | 1DP2DE024365-01 | Christian P Petersen |

The funders had no role in study design, data collection and interpretation, or the decision to submit the work for publication.

### Author contributions

Eric M Hill, Conceptualization, Investigation, Writing—original draft, Writing—review and editing; Christian P Petersen, Conceptualization, Data curation, Supervision, Funding acquisition, Writing—original draft, Writing—review and editing

### Author ORCIDs

Eric M Hill http://orcid.org/0000-0003-1426-2573
Christian P Petersen http://orcid.org/0000-0001-7552-6865

### Decision letter and Author response

Decision letter https://doi.org/10.7554/eLife.33680.023
Author response https://doi.org/10.7554/eLife.33680.024

## Additional files

### Supplementary files
• Transparent reporting form
DOI: https://doi.org/10.7554/eLife.33680.021

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
