## [Decision Letter]

Thank you for submitting your article "Patterning alteration uncouples regeneration from homeostatic tissue maintenance in planarians" for consideration by *eLife*. Your article has been reviewed by three peer reviewers, and the evaluation has been overseen by a Reviewing Editor and Marianne Bronner as the Senior Editor. The following individuals involved in review of your submission have agreed to reveal their identity: Bret J Pearson (Reviewer #2) and Alejandro Sánchez Alvarado (Reviewing Editor).

The reviewers have discussed the reviews with one another and the Reviewing Editor has drafted this decision to help you prepare a revised submission.

Summary:

By taking advantage of the eye duplication phenotype triggered by *notum(RNAi)*, Hill and Petersen examine mechanisms that coordinate organ patterning, homeostasis, and regeneration in planarians. The authors show that ectopic eyes contain correct cell types, project to the brain, and are also maintained homeostatically. However, if the original eyes are specifically removed, they fail to regenerate. Further, the authors show that ectopic eyes do not cause ectopic increases of eye progenitors, and existing eyes will compete for progenitor cells during homeostasis. Following analysis of duplicated anterior eyes, the authors used *wnt11-6* and *fzd5/8-4* RNAi conditions to shift ectopic eyes to appear posterior to the original eyes. In this case, similar to above, the original eyes will not regenerate. Additionally, the authors show similar properties to be displayed by ectopic pharynges.

This study is novel and imaginative. The experiments are well-designed, technically sound, and stainings and imaging are of high quality. The main finding is remarkable, and it is well supported by experiments using multiple morphogens and organ structures to document this effect.

Essential revisions:

Before this manuscript can be considered further, the following concerns must be addressed:

1) Incorporation of progenitors: In both cases where BrdU staining is shown, the data is not convincing. Quantification over time, together with DAPI, should be included for BrdU staining to support the claim made by authors that both sets of photoreceptors are indeed maintained. As the fundamental drivers of homeostasis and regeneration, understanding the relative contributions of neoblasts to regenerating photoreceptors is a necessary component of the argument that the authors attempt to make here. Hence, quantitative assessment of the number of progenitors and the incorporation of progenitors into the two sets of eyes under the various conditions proposed (homeostasis *notum(RNAi)* (Figure 2), *notum(RNAi)* with eye scrape (Figure 4), *notum(RNAi)* with eye scrape and neck wound (Bret), would go a long way in addressing this concern. This would involve BrdU and *ovo* labelling and tracking the number of progenitors as well as where they incorporate.

2) It is not clear how *notum(RNAi)* animals maintain 2 sets of eyes with the same number of *ovo* cells as a regular animal. Does that mean that both sets of eyes are smaller than the eyes of a wildtype animal? And how can it be that in Figure 4 the growth of the posterior eye while also regenerating an anterior eye (bottom panel) is increased compared to the contralateral eye – and that this is even to the same extent it seems as the increased growth of the anterior eye when the posterior eye is permanently eliminated (top panel). It seems that this would only be possible if the total number of *ovo* cells was also increased in the right side of the bottom panel, but it is thought that such eye resections don't elicit a missing tissue response. LoCascio et al. showed that eye scratch plus neck wounding will increase eye progenitors. In the background of *notum* RNAi, the authors should perform original-eye scratches with neck wounding to see whether this increase in eye progenitors would be enough to maintain the original eye. It would be helpful to further clarify this observation.

3) While the authors did a nice job with their ectopic posterior eye experiments, all reviewers would like to see a set of experiments done with *nou-darake* RNAi, which will generate ectopic eyes much further posterior to original eyes. It would be informative to see a condition where original and ectopic eyes can both be regenerated after a scratch. It would also help the model to show that if an eye field can be greatly removed, there is no interference between sets of eyes that are so close as in *notum* RNAi.

4) The authors claim that the photoreceptors are shifted posteriorly to positional control genes (PCGs). However, it also seems plausible that long-term RNAi of *notum* causes broad axial patterning changes and fundamentally alters progenitor behavior, since it still remains mysterious how these PCGs communicate with neoblasts to control regenerative potential. As shown in Figure 3, the pre-existing eyes shift posteriorly, yet the number and domain of *cintillo*+ cells decreases, and the relative positioning of PRs to *cintillo*+ cells is changed significantly. This suggests pleiotropic changes in the animals that may influence progenitor behavior. The experiments in Figure 6, which are done in wild-type animals, circumvent this issue, but the authors fail to merge it mechanistically beyond a correlation. Linking the function of patterning molecules with neoblast behavior during regeneration or homeostasis would be a valuable contribution.

5) The authors propose that after the initial stage of regeneration the rescaled body axes direct the morphallaxis – or more specifically, as in Figure 7, the gradual repositioning of the pre-existing eye to the new position which is proportional to the size of the new body. Intuitively we know that this must be true, because if repositioning would not happen, the stereotypic body plan of a planarian would have been long lost. The question then however remains why this does not happen to the mis-positioned pre-existing eyes in *notum(RNAi)* or any of the other conditions that cause a shift in the body axes. It seems from their data that other organs such as the brain do rescale to match the new positional information, so what is it about the eyes and the pharynx that makes them behave differently and create a new second structure in the correct anatomical position instead of just moving the old one? This is not due to the stronger role of morphallaxis during regeneration (as in Figure 7) versus homeostasis, because in Figure 1—figure supplement 2 authors show that the exact procedure of Figure 7 in a *notum(RNAi)* animal leads to double eyes. Their data concerning the regeneration of the eye only in the new correct position is convincing, but this last piece of the puzzle seems to not fit entirely.

---

## [Author Response]

Essential revisions:Before this manuscript can be considered further, the following concerns must be addressed:1) Incorporation of progenitors: In both cases where BrdU staining is shown, the data is not convincing. Quantification over time, together with DAPI, should be included for BrdU staining to support the claim made by authors that both sets of photoreceptors are indeed maintained. As the fundamental drivers of homeostasis and regeneration, understanding the relative contributions of neoblasts to regenerating photoreceptors is a necessary component of the argument that the authors attempt to make here. Hence, quantitative assessment of the number of progenitors and the incorporation of progenitors into the two sets of eyes under the various conditions proposed (homeostasis notum(RNAi) (Figure 2), notum(RNAi) with eye scrape (Figure 4), notum(RNAi) with eye scrape and neck wound (Bret), would go a long way in addressing this concern. This would involve BrdU and ovo labelling and tracking the number of progenitors as well as where they incorporate.

We thank the reviewers for this insightful suggestion. A central claim of our work is that nonregenerative eyes produced from pattern alteration are homeostatically maintained, and we agree that the BrdU incorporations experiments are critical for this argument. In particular, in the first submission we had initially scored the eyes has either having or not having any BrdU^+^ cells, so the reviewers raise a valid concern over whether the nonregenerative eyes are truly homeostatically maintained versus incorporating new eye cells only rarely and to a lower extent than regenerative eyes. We now include BrdU analysis of eye cell numbers in each eye over time (Figure 2). We find that *notum* RNAi regenerative and nonregenerative eyes acquire approximately equal numbers of new eye cells, indicating homeostatic maintenance occurs as robustly in both types of eyes. We also find that *notum* RNAi does not alter numbers of *ovo*+ progenitor cells in uninjured animals (Figure 4) or in animals responding to eye resection (Figure 4—figure supplement 2). These observations further support the complementary evidence we give for the long-term maintenance of nonregenerative eyes by tracking them over an extended time (over 3x the time required for complete turnover) and finding that their persistence requires the eye differentiation factor *ovo*. Therefore, long-term persistence occurs in conjunction with new eye cell differentiation and requires eye cell differentiation, strongly supporting the model that they are actively homeostatically maintained.

2) It is not clear how notum(RNAi) animals maintain 2 sets of eyes with the same number of ovo cells as a regular animal. Does that mean that both sets of eyes are smaller than the eyes of a wildtype animal?

We thank the reviewers for helping to clarify these results. First, while analyzing BrdU labeled uninjured animals, we found that the rate of eye cell incorporation is indeed about half for each of the four *notum* RNAi eyes compared to control RNAi eyes and that the total rates of eye cell production are similar between the two conditions (Figure 2). Uninjured animals correspondingly have similar numbers of *ovo*+ progenitors (Figure 4). We counted eye cells and found that, as the reviewers predicted, both sets of *notum* RNAi eyes are smaller than control eyes, but the total eye cell number is similar (Figure 4).

And how can it be that in Figure 4 the growth of the posterior eye while also regenerating an anterior eye (bottom panel) is increased compared to the contralateral eye – and that this is even to the same extent it seems as the increased growth of the anterior eye when the posterior eye is permanently eliminated (top panel). It seems that this would only be possible if the total number of ovo cells was also increased in the right side of the bottom panel, but it is thought that such eye resections don't elicit a missing tissue response.

For the experiments in Figure 4 (formerly 4B), both eye removal treatments resulted in L/R eye size asymmetry. The graphs are scaled to show the effect (which is significant in each case), though the size of the effect is actually greater after removal of the nonregenerative posterior eye (causing reallocation of ~40 eye cells to the anterior photoreceptor) compared to removal of a regenerative eye (causing reallocation of ~10-20 cells to the posterior photoreceptor). Our interpretation of this difference in effect size is that the posterior eye cannot regenerate, resulting in an additional number of progenitors that can be re-allocated to the anterior eye. We suggest that an explanation for the behavior of the system after removal of the anterior eye is that while this eye is early in its regeneration process there is effectively no anterior eye for several days during which time the posterior eye can competitively acquire additional progenitors, though apparently not to the exclusion of the anterior eye’s regenerative ability. We now provide further experiments now showing that the overall numbers of eye cells and overall rates of eye cell differentiation are not altered by *notum* RNAi (Figure 2, Figure 4). Furthermore, data from LoCascio 2017 argue strongly that small injuries to remove eyes do not further alter the global rates of eye differentiation. Therefore, our interpretation of the outcomes of Figure 4 are that eyes compete to acquire an existing pool of progenitors so this competitive landscape is modified by eye removal.

LoCascio et al. showed that eye scratch plus neck wounding will increase eye progenitors. In the background of notum RNAi, the authors should perform original-eye scratches with neck wounding to see whether this increase in eye progenitors would be enough to maintain the original eye. It would be helpful to further clarify this observation.

We also tried this interesting suggestion to determine whether a nearby injury and its known ability to trigger production of extra eye progenitors would be enough to allow regeneration of the posterior *notum* RNAi eyes that are otherwise nonregenerative. We found that wedge amputations posterior to the eyes did not enable posterior eye regeneration (Figure 4—figure supplement 1), and behaved identically to animals with no neck wounding. This is consistent with our other analysis showing that posterior eyes have access to eye progenitors for homeostatic maintenance (BrdU staining in Figure 2 and *ovo*+ cell distributions in Figure 4), suggesting that the lack of posterior eye regenerative ability is not likely due to insufficient eye progenitors.

3) While the authors did a nice job with their ectopic posterior eye experiments, all reviewers would like to see a set of experiments done with nou-darake RNAi, which will generate ectopic eyes much further posterior to original eyes. It would be informative to see a condition where original and ectopic eyes can both be regenerated after a scratch. It would also help the model to show that if an eye field can be greatly removed, there is no interference between sets of eyes that are so close as in notum RNAi.

We tested eye regeneration in *nou darake* RNAi animals that form ectopic posterior eyes (Figure 5—figure supplement 3). In *Schmidtea mediterranea* we have found that the brain expansion phenotype in these animals is highly penetrant whereas ectopic eyes occur at a lower penetrance (~50%) and appeared in the neck region. We performed resections on these animals using the same protocol we had used for *notum* RNAi and *wnt11-6/fzd45-8* RNAi and found that, surprisingly, pre-existing *ndk(RNAi)* eyes were capable of regeneration while ectopic *ndk(RNAi)* eyes did not appear capable of regeneration. This would seem to suggest that *ndk* and *wnt11-6* could act through distinct mechanisms to pattern the anterior and/or restrict brain/eye regionalization. One possibility is that *ndk* factors influence eye progenitor differentiation while *wnt11-6* influences the target location of regeneration. We feel that this is an interesting distinction that would merit further analysis in an independent study designed to study these two pathways but is less related to the goal of the current study in showing that homeostatic tissue maintenance can occur at a site distinct from regeneration. We include this data here (Figure 5—figure supplement 3) to emphasize that modifying axis composition itself does not necessarily alter the site of regeneration.

To address the issue of eye regenerative ability in a context with eyes displaced at a greater distance, we also used longer-term *notum* RNAi and surgery to form 6-eyed animals in which two original eyes are more posterior than two sets of ectopic eyes. In this case, only the most anterior eyes were able to regenerate while the more distant posterior eyes were incapable of regeneration (Figure 1—figure supplement 3).

To address the issue of interference, we also conducted experiments in 4-eyed *notum* RNAi animals in which both anterior and posterior eyes were removed. Only the anterior eyes regenerate in this context, indicating the lack of posterior eye regeneration is not due to the presence of the anterior eye (Figure 1—figure supplement 2, top panels).

4) The authors claim that the photoreceptors are shifted posteriorly to positional control genes (PCGs). However, it also seems plausible that long-term RNAi of notum causes broad axial patterning changes and fundamentally alters progenitor behavior, since it still remains mysterious how these PCGs communicate with neoblasts to control regenerative potential. As shown in Figure 3, the pre-existing eyes shift posteriorly, yet the number and domain of cintillo+ cells decreases, and the relative positioning of PRs to cintillo+ cells is changed significantly. This suggests pleiotropic changes in the animals that may influence progenitor behavior. The experiments in Figure 6, which are done in wild-type animals, circumvent this issue, but the authors fail to merge it mechanistically beyond a correlation. Linking the function of patterning molecules with neoblast behavior during regeneration or homeostasis would be a valuable contribution.

We found that nonregenerative photoreceptors do indeed have abnormal locations with respect to anterior positional control gene expression domains (Figure 3). *Notum* itself is a positional control gene and we agree that long-term *notum* RNAi produces changes to axis composition that are likely mediated by modulation of other Wnts acting as positional control genes. We showed previously that the *notum* RNAi ectopic eye phenotype is suppressed by co-inhibition of *wnt11-6*, as were effects of the brain size due to *wnt11-6* RNAi (Hill and Petersen, 2015), suggesting that these two factors oppositely pattern both aspects of the anterior. Our impression of the *notum* RNAi phenotype is that it seems to involve some anterior growth that includes brain tissue so that to an extent original eyes do not move posterior but rather became posteriorly shifted due to this broader anterior axis extension. Eyes always seem to track a particular location with respect to the planarian brain, but because there are no known phenotypes of brain absence but eye presence, there is not yet a way to fully clarify the dependencies of these two effects. We now describe in interpreting these experiments that it is possible that the brain size/position dictates the site of new eye cell regeneration or alternatively that a single process can control both properties (Results, sixth paragraph). It is an interesting suggestion that *notum(RNAi)* modifies progenitor behavior. We do not see evidence of strong effects on eye progenitor numbers, distribution, or overall eye differentiation ability after *notum* RNAi. The question of how the positional control gene network interfaces with neoblast activities and the sites of regeneration is a very interesting topic that we hope to pursue in future research.

5) The authors propose that after the initial stage of regeneration the rescaled body axes direct the morphallaxis – or more specifically, as in Figure 7, the gradual repositioning of the pre-existing eye to the new position which is proportional to the size of the new body. Intuitively we know that this must be true, because if repositioning would not happen, the stereotypic body plan of a planarian would have been long lost. The question then however remains why this does not happen to the mis-positioned pre-existing eyes in notum(RNAi) or any of the other conditions that cause a shift in the body axes. It seems from their data that other organs such as the brain do rescale to match the new positional information, so what is it about the eyes and the pharynx that makes them behave differently and create a new second structure in the correct anatomical position instead of just moving the old one? This is not due to the stronger role of morphallaxis during regeneration (as in Figure 7) versus homeostasis, because in Figure 1—figure supplement 2 authors show that the exact procedure of Figure 7 in a notum(RNAi) animal leads to double eyes. Their data concerning the regeneration of the eye only in the new correct position is convincing, but this last piece of the puzzle seems to not fit entirely.

All of the organs we have investigated so far scale with body size and so any regionalized tissues (such as the brain, eyes, pharynx) are subject to size and position control by morphallaxis/remodeling. In general these remodeling events have not been observed to generate duplicated tissue, as would be predicted in a model in which positional information is restored rapidly, followed by synthesis of correctly sized/positioned organs and degradation of old mispositioned organs. In the case of the eye, we only rarely observed this duplication in normal animals. A similar process could occur within the brain, but because this tissue is more spread out than an eye, it may be more difficult to observe. It is also possible that other organs will use different strategies for tissue remodeling, and we now mention both possibilities (Discussion, second paragraph). Alternatively, differences in rates of production or also in the ability for neoblasts and their progeny to respond to A/P positional information could account for these differences. Future discoveries of brain-specific progenitor cells and also their responsiveness to patterning gene signaling will be useful in clarifying these models. However, we feel that a compelling and potentially generalizable model from our analysis of the eye and pharynx is that the flexibility of tissue homeostasis with respect to position helps buffering against the short-term changes to positional information during remodeling. Though our model remains incomplete, our hope is that it will be a useful framework for future studies aiming to understand these aspects of whole-body regeneration.